# Fibronectin Promotes Cell Growth and Migration in Human Renal Cell Carcinoma Cells

**DOI:** 10.3390/ijms20112792

**Published:** 2019-06-07

**Authors:** Yen-Chuan Ou, Jian-Ri Li, Jiaan-Der Wang, Cheng-Yi Chang, Chih-Cheng Wu, Wen-Ying Chen, Yu-Hsiang Kuan, Su-Lan Liao, Hsi-Chi Lu, Chun-Jung Chen

**Affiliations:** 1Department of Urology, Tungs’ Taichung MetroHarbor Hospital, Taichung 435, Taiwan; ycou228@gmail.com; 2Division of Urology, Taichung Veterans General Hospital, Taichung 407, Taiwan; fisherfishli@yahoo.com.tw; 3Department of Children’s Medical center, Taichung Veterans General Hospital, Taichung 407, Taiwan; wangjiaander@gmail.com; 4Department of Surgery, Feng Yuan Hospital, Taichung 420, Taiwan; c.y.chang.ns@gmail.com; 5Department of Anesthesiology, Taichung Veterans General Hospital, Taichung 407, Taiwan; chihcheng.wu@gmail.com; 6Department of Financial Engineering, Providence University, Taichung 433, Taiwan; 7Department of Data Science and Big Data Analytics, Providence University, Taichung 433, Taiwan; 8Department of Veterinary Medicine, National Chung Hsing University, Taichung 402, Taiwan; wychen@dragon.nchu.edu.tw; 9Department of Pharmacology, Chung Shan Medical University, Taichung 402, Taiwan; kuanyh001@gmail.com; 10Department of Medical Research, Taichung Veterans General Hospital, Taichung 407, Taiwan; slliao@vghtc.gov.tw; 11Food Science Department and Graduate Institute, Tunghai University, Taichung 407, Taiwan; hclu@thu.edu.tw; 12Department of Medical Laboratory Science and Biotechnology, China Medical University, Taichung 404, Taiwan

**Keywords:** extracellular matrix, fibronectin, motility, renal cell carcinoma

## Abstract

The prognostic and therapeutic values of fibronectin have been reported in patients with renal cell carcinoma (RCC). However, the underlying mechanisms of malignancy in RCC are not completely understood. We found that silencing of fibronectin expression attenuated human RCC 786-O and Caki-1 cell growth and migration. Silencing of potential fibronectin receptor integrin α5 and integrin β1 decreased 786-O cell ability in movement and chemotactic migration. Biochemical examination revealed a reduction of cyclin D1 and vimentin expression, transforming growth factor-β1 (TGF-β1) production, as well as Src and Smad phosphorylation in fibronectin-silenced 786-O and Caki-1 cells. Pharmacological inhibition of Src decreased 786-O cell growth and migration accompanied by a reduction of cyclin D1, fibronectin, vimentin, and TGF-β1 expression, as well as Src and Smad phosphorylation. In 786-O cells, higher activities in cell growth and migration than in Caki-1 cells were noted, along with elevated fibronectin and TGF-β1 expression. The additions of exogenous fibronectin and TGF-β1 promoted Caki-1 cell growth and migration, and increased cyclin D1, fibronectin, vimentin, and TGF-β1 expression, as well as Src and Smad phosphorylation. These findings highlight the role of fibronectin in RCC cell growth and migration involving Src and TGF-β1 signaling.

## 1. Introduction

Renal cell carcinoma (RCC) is a lethal malignancy that originates in the kidney and is characterized by highly vascularized and metastatic features. Among the histological subtypes of this cancer, clear cell RCC is the most common type. Despite advances in early detection of small renal masses and the effectiveness of radical nephrectomy, approximately one third of patients with RCC have already developed metastases at diagnosis, eventually leading to metastatic lesions during the progression of this disease. Once metastasis has occurred, the five-year survival rate of patients with advanced RCC is unsatisfactory [1,2]. Clinically, refractoriness to adjuvant therapeutic modalities, including targeted therapy, radiotherapy, chemotherapy, hormonal therapy, and immunotherapy, further worsens the poor prognosis [3,4]. Therefore, a better understanding of the molecular basis of metastases and resistance is of the utmost importance as it may permit the development of a clinically feasible therapeutic strategy against RCC.

Extracellular matrix forms a basic scaffold for the normal physiology of cells. The interaction between cell and extracellular matrix also plays a central role in the initiation and progression of malignancies. Among the extracellular matrix proteins identified, fibronectin is able to activate a series of signal transduction pathways by interacting with various receptors, thereby regulating cellular processes, including adhesion, migration, invasion, proliferation, differentiation, and morphogenesis. Accumulating evidence has indicated its carcinogenic and malignant role in several malignancies [5,6,7,8,9]. In RCC, tissue and plasma levels of fibronectin are elevated in the advanced stages, and thus fibronectin appears to be a promising biomarker for therapeutic responses [8,10,11,12,13,14,15,16,17]. Although data from clinical and experimental studies highlight the role of fibronectin in RCC prognosis and acquisition of RCC malignancy, the underlying molecular and biochemical basis of fibronectin in promoting malignant progression of RCC remain incompletely understood.

Fibronectin is a secreted extracellular matrix component and a member of the mesenchymal protein family, and, therefore, its biological activity is believed to be mainly governed via a process known as epithelial–mesenchymal transition (EMT). However, the engagement of fibronectin with integrin receptors might also actively initiate and transduce distinct postreceptor intracellular signaling, thereby contributing to dynamic regulation of cell adhesion, migration, invasion, and proliferation. Evidence indicates that focal adhesion kinase (FAK), Src, extracellular signal-regulated kinase (ERK), Akt, and transforming growth factor-β1 (TGF-β1) are concurrent or concomitant effectors of postreceptor actions of fibronectin [9,18,19,20,21]. However, our current understanding of cellular events in fibronectin-promoted RCC cell growth and migration is limited. Since fibronectin overexpression in RCC has been strongly associated with advanced clinical stages, metastases, and poor survival [8,14,16,17], the aim of the current study was to investigate the intracellular signaling events critical to fibronectin-promoted cell growth and migration.

## 2. Results

### 2.1. Fibronectin Silencing Decreased Cell Growth and Migration

The importance of fibronectin in RCC cell biology was first investigated by RNA interfering (RNAi) strategy. Fibronectin silencing by siRNA caused a reduction of fibronectin protein expression in human RCC 786-O cells (Figure 1A) and led to decreased cell growth (Figure 1B) and long-term clonogenic cell growth (Figure 1C) when compared with the corresponding control. Controlled 786-O cells started to adhere to dishes and protrude cell processes as early as 1 h after seeding. In contrast, most fibronectin-silenced 786-O cells still displayed bright and rounded morphology at the same time (Figure 1D). Fibronectin-silenced 786-O cells showed decreased ability in wound healing (Figure 1E) and chemotactic migration towards 10% fetal bovine serum (FBS) (Figure 1F) than control cells. These findings indicate that silencing of endogenous fibronectin expression has a negative effect on RCC cell growth and migration.

### 2.2. Integrin α5 and Integrin β1 Silencing Alleviated Fibronectin Effects

To further investigate the effects of fibronectin on RCC cell growth and migration, 786-O cells were seeded onto fibronectin-coated cultured plates and Transwell inserts. The presence of exogenous fibronectin promoted cell growth (Figure 2A) and chemotactic migration towards 10% FBS (Figure 2B). When compared with vehicle control, 786-O cells also displayed higher chemotactic migration towards fibronectin (Figure 2C). Since dimeric integrin α5 and integrin β1 are representative cell surface receptors of fibronectin [9,12], their potential roles were investigated. Antibody neutralization study revealed a potential involvement of integrin α5 and integrin β1 in fibronectin-mediated cell migration (Figure 2D). Parallel studies revealed that silencing of endogenous integrin α5 and integrin β1 expression (Figure 2E) decreased cell ability in wound healing (Figure 2F), chemotactic migration towards 10% FBS (Figure 2G), fibronectin-increased migration (Figure 2H), and chemotactic migration towards fibronectin (Figure 2I). Our findings indicate that fibronectin and its dimeric receptor integrin α5/integrin β1 play a role in RCC cell growth and migration.

### 2.3. Fibronectin Silencing Decreased Intracellular Src Signaling

The aforementioned findings imply that fibronectin plays a substantial role in RCC cell growth and migration. Since FAK, Src, ERK, and Akt have been implicated in transducing signals of fibronectin to induce distinct cellular activities [9,18,19,20,21], the expression of these molecules and accompanying effectors were examined. An apparent reduction of Src phosphorylation was noted in fibronectin-silenced 786-O cells, whereas the changes of integrin α5, integrin β1, FAK phosphorylation, ERK phosphorylation, and Akt phosphorylation were not significant (Figure 3A). Additionally, fibronectin-silenced 786-O cells decreased expression of cell growth- and migration-associated molecules, including cyclin D1, vimentin, Smad phosphorylation (Figure 3A), and TGF-β1 (Figure 3B). The role and importance of Src signaling in RCC cell growth and migration were evaluated by pharmacological inhibition with PP2 inhibitor [9]. A reduction of Src phosphorylation, cyclin D1, fibronectin, vimentin, and Smad phosphorylation was observed in PP2-treated 786-O cells (Figure 3C). Furthermore, PP2-treated 786-O cells showed decreased cell growth (Figure 3D), chemotactic migration towards 10% FBS (Figure 3E), and TGF-β1 production (Figure 3F). That is, the reduction of Src signaling had a substantial effect on the cellular activities of fibronectin-silenced 786-O cells.

### 2.4. Fibronectin Silencing Decreased Cell Growth and Migration in Caki-1 Cells

To further demonstrate the cell growth and migration inhibitory effects of fibronectin silencing on RCC cells, human RCC Caki-1 cells were also investigated. As with 786-O cells, fibronectin silencing caused a reduction in Src phosphorylation, cyclin D1, vimentin, Smad phosphorylation (Figure 4A), cell growth (Figure 4B), chemotactic migration towards 10% FBS (Figure 4C), and TGF-β1 production (Figure 4D) in Caki-1 cells. The findings indicate that the axis of fibronectin and Src is a regulatory mechanism of cell growth and migration in RCC cells.

### 2.5. Cell Growth and Migration Potential of RCC Cells Were Associated with Fibronectin

To investigate whether a correlation existed between fibronectin level and cell growth/migration potential, a biochemical comparison was conducted in 786-O and Caki-1 cells. In subconfluent cells, 786-O cells expressed higher amounts of cytoplasmic and extracellular fibronectin along with integrin β1 compared with Caki-1 cells. However, the level of integrin α5 in 786-O cells was lower than in Caki-1 cells (Figure 5A). In parallel, 786-O cells showed more of an increased ability than Caki-1 cells in cell growth (Figure 5B), wound healing (Figure 5C), chemotactic migration towards 10% FBS (Figure 5D), and TGF-β1 production (Figure 5E). These results suggest that fibronectin might have value as a candidate molecule for determining cell growth and migration among RCC cells.

### 2.6. Fibronectin Increased Src and TGFβ-1 Signaling

As with 786-O cells, exogenous fibronectin promoted cell growth (Figure 6A), chemotactic migration towards 10% FBS (Figure 6B), chemotactic migration towards fibronectin (Figure 6C), and TGF-β1 production (Figure 6D) in Caki-1 cells. Neutralization of membrane integrin α5 and integrin β1 receptor decreased cell ability in chemotactic migration towards fibronectin (Figure 6C). Biochemically, exogenous addition of fibronectin increased Src phosphorylation, Smad phosphorylation, cyclin D1, and vimentin (Figure 6E) in Caki-1 cells. In order to observe any potential bidirectional causal effects, Caki-1 cells were treated with TGF- β1. TGF-β1 promoted cell growth (Figure 7A) and chemotactic migration towards 10% FBS (Figure 7B). The enhanced cellular activities by TGF-β1 were accompanied by increased Smad phosphorylation, Src phosphorylation, cyclin D1, fibronectin, and vimentin (Figure 7C). The findings suggest a potential interaction among fibronectin, Src, and TGF-β1 in RCC cells.

### 2.7. Hypoxia Increased RCC Cell Migration

In addition to normoxic conditions, cell migration was evaluated under a hypoxic condition. Compared with normoxic condition, hypoxia for 8 h was able to promote 786-O cell migration towards 10% FBS (Figure 8A). Western blotting study revealed that the levels of fibronectin, integrin α5, integrin β1, Src phosphorylation, and Smad phosphorylation were not changed by hypoxia. Instead, the levels of FAK phosphorylation and Akt phosphorylation were elevated by hypoxia (Figure 8B). As with 786-O cells, hypoxia promoted Caki-1 cell migration towards 10% FBS (Figure 8C), increased FAK phosphorylation and Akt phosphorylation, but left fibronectin, integrin α5, integrin β1, Src phosphorylation, and Smad phosphorylation unchanged (Figure 8D). These findings indicate that hypoxia is a driving force of RCC cell migration without significant effect on fibronectin expression.

## 3. Discussion

The interaction between cancer cells and extracellular compartments plays a pivotal role in tumor progression. In the noncellular microenvironment, extracellular matrix proteins such as fibronectin, laminin, osteopontin, collagen, glycosaminoglycans, and proteoglycan are of considerable importance. Fibronectin is upregulated in RCC and is therefore of prognostic value. Moreover, it has been shown to be a promising biomarker for therapeutic responses [8,10,11,12,13,14,15,16,17]. In this study, manipulation of cellular fibronectin expression by RNAi and exogenous additions altered cellular activities and intracellular signaling in RCC 786-O and Caki-1 cells. Our findings demonstrate that fibronectin exerted a positive effect with respect to RCC cell growth and migration involving membrane-spanning receptors integrin α5 and integrin β1. Data from biochemical studies revealed a positive correlation between fibronectin and the protein expression or protein phosphorylation of cyclin D1, vimentin, TGF-β1, Src, and Smad. Interaction among fibronectin, Src, and TGF-β1/Smad related to RCC cellular activities and intracellular signaling was further demonstrated by the results of experiments involving Src inhibitor- and TGF-β1-treated cells. Although RCC cell migration was promoted by hypoxia, the level of fibronectin expression was not significantly changed by hypoxia. Therefore, our findings highlight a possible mechanism for the regulation of fibronectin expression and its promotion of cell growth and migration in RCC. However, its cellular activities vary and depend on microenvironments.

The effects of extracellular matrix proteins on cellular activities are multifactorial. Transmembrane protein receptors play a crucial role in transducing extracellular signals after ligand and receptor engagement. It has been demonstrated that tissue transglutaminase, osteopontin, integrin α5, integrin αv, integrin β1, and integrin β3 bind to and are activated by fibronectin, and have been implicated in tumor cell proliferation, adhesion, migration, invasion, stemness, and resistance in a number of cancers, including RCC. The roles and importance of dimeric integrin receptor subunits vary and depend on cell types and microenvironments [18,19,21,22]. Dimeric integrin α5 and integrin β1 are common cell surface receptors of fibronectin [9,12,18]. Here, we primarily silenced the expression of integrin α5 and integrin β1 and found that integrin α5 or integrin β1 silencing alone decreased spontaneous cellular abilities in wound healing and chemotactic migration. Moreover, the silenced cells displayed decreased chemotactic migration towards fibronectin. The potential involvement of integrin α5 and integrin β1 receptor in RCC cell migration towards fibronectin was further demonstrated by data of antibody neutralization studies. Likewise, our findings suggest a possible role of integrin α5 and integrin β1 subunits in RCC cellular activities and in the mechanistic action of fibronectin. Intriguingly, neutralization of integrin β1 appeared to show better inhibitory effect than integrin α5 in chemotactic migration towards fibronectin. However, both neutralizations failed to cause a complete inhibition. Current findings remind us to keep in mind that additional integrin receptor subunits may also have roles in RCC chemotactic migration towards fibronectin. It should be noted that scratch assay or wound healing assay consists of two cellular activities, cell proliferation and migration. Although the wound healing assay was conducted by maintaining the cells in 0.5% FBS to minimize cell proliferation, the effect of cell proliferation could not be totally excluded. Since the precise mechanism of intracellular signaling in integrin-silenced cells and the potential involvement of other receptors were not addressed in this study, the detailed action mechanisms of corresponding fibronectin receptors should be further investigated.

Fibronectin has been demonstrated to act as an extracellular cue, which facilitates the coordination of the functional and structural dynamics of cells through several intermediate effectors, including FAK, Src, ERK, and Akt, leading to the induction of transcriptional processes and cytoskeletal reorganization [9,18,19,20,21]. Consistent with relevant studies, our biochemical and pharmacological findings also demonstrated the critical role of Src in fibronectin-mediated intracellular signaling and cellular activities. Unexpectedly, the accompanying change in FAK, ERK, and Akt phosphorylation was not detected in fibronectin-silenced cells. Evidence indicates the actions and intracellular signaling of fibronectin could be complicated and may rely on a diverse range of microenvironments [21]. Our investigation was limited to the biochemical measurement of protein expression and phosphorylation in fibronectin-silenced cells 48 h after siRNA transfection. Although the timing might be a potential cause of the discrepancy, we did not design further advanced experiments to explore this issue.

The findings presented herein showed that knockdown of fibronectin expression in RCC cells caused a subsequent reduction in cell growth- and migration-associated molecules, including cyclin D1, vimentin, and TGF-β1, together with decreased Src and Smad phosphorylation. The addition of exogenous fibronectin reversed the abovementioned effects in terms of cell growth, migration, cyclin D1 expression, vimentin expression, TGF-β1 production, Src phosphorylation, and Smad phosphorylation. In its role as a master signaling molecule, Src is capable of converging diverse signals for the purpose of initiating the transcriptional program. The expression of cyclin D1, TGF-β1, vimentin, and fibronectin is under the control of Src [9,23]. The role and importance of Src in RCC cell growth, migration, and the effects of fibronectin were further demonstrated by its pharmacological inhibition. As with fibronectin silencing, pharmacological inhibition of Src caused a reduction in cell growth, migration, TGF-β1 production, cyclin D1/vimentin/fibronectin expression, and Src/Smad phosphorylation. Intriguingly, increased cell growth, migration, cyclin D1/vimentin/fibronectin expression, and Src/Smad phosphorylation were observed in TGF-β1-treated cells. In addition to being a mesenchymal protein, fibronectin may operate through the integrin α5/Src pathway to establish EMT [24,25]. Moreover, Src plays a determinant role in TGF-β1-induced EMT [26]. Paralleling the results of experiments with fibronectin, the expression and activity of vimentin, cyclin D1, TGF-β1, Smad, and Src are elevated in tumor tissues of RCC and show prognostic potential [27,28,29,30,31]. Therefore, regardless of the upstream signals, the aforementioned findings and our results suggest that TGF-β1, Src, and fibronectin could work in concert to shape a mesenchymal-like morphology and predispose RCC cells to cell growth and migration, involving induction of cyclin D1, vimentin, and other unidentified molecules.

786-O and Caki-1 are two common cell lines of human RCC with distinct genetic backgrounds [11]. We found that 786-O cells showed higher potential in cell growth, wound healing, and chemotactic migration than Caki-1 cells, which reflects our findings showing elevated fibronectin, integrin β1, and TGF-β1 expression. However, the expression level of integrin α5 in 786-O cells was lower than that in Caki-1 cells. Data of biochemical comparison between 786-O and Caki-1 cells suggested that integrin β1 might be one potential cause in explaining discrepancy between their migration ability because of a better effect produced by integrin β1 neutralization and higher expression of integrin β1 in 786-O cells. Han et al. [11] identified a higher expression of estrogen receptor β in 786-O cells than in Caki-1 and A498 cells, a phenomenon that may play a role in promoting RCC cell invasion. Our pilot findings also revealed an elevated expression of Yes-associated protein (YAP1), Runt-related transcription factor 2 (*RUNX2*), β-catenin, and Akt phosphorylation in 786-O cells compared with Caki-1 cells (data not shown). Although the findings of the current study suggest a role of integrin α5/integrin β1 and the existence of interaction among fibronectin, Src, and TGF-β1 signaling, and that these components play roles in the development of RCC, the complicated phenomena remind us that the roles of other uncharacterized molecules should not be ignored.

Hypoxia represents an alternative factor in regulating the progression of RCC [32]. The actions and signaling of fibronectin, Src, TGF-β1/Smad, FAK, and Akt have substantial roles in hypoxia-mediated cell migration [9]. As with other cancerous cells [9], hypoxia (8 h) significantly promoted RCC cell migration. Under normoxic conditions, the aforementioned findings indicated that RCC cell migration was associated with increased fibronectin expression and accompanying Src and TGF-β1/Smad signaling. Intriguingly, hypoxia-promoted cell migration was not accompanied by altered fibronectin, integrin α5, integrin β1, Src phosphorylation, and Smad phosphorylation. Instead, the levels of FAK phosphorylation and Akt phosphorylation were elevated by hypoxia. Bluyssen et al. [33] found that the expression of fibronectin was independent of hypoxia in 786-O cells. Their findings might partially explain why the expression of fibronectin in RCC cells remained unchanged following exposure to hypoxia. The action mechanisms of hypoxia-mediated cell migration are multifactorial. The detailed biochemical changes and cellular activities in mediating hypoxia-promoted RCC cell migration require further investigation.

The regulation of cell proliferation and migration is multifactorial. Studies have highlighted the crucial role of fibronectin in RCC cell proliferation and motility, and have demonstrated its prognostic value [8,10,11,12,13,14,15,16,17]. Using 786-O and Caki-1 cells with fibronectin silencing, the results of our study further emphasize the importance and crucial role of fibronectin, Src, and TGF-β1/Smad in the regulation of RCC cell growth and migration. Fibronectin, Src, and TGF-β1/Smad may form an interactive network for mutual expression, thereby predisposing cells to growth and migration. Although there are still many candidate molecules that have yet to be investigated, the biochemical and molecular evidence presented herein may serve to advance the current understanding of the pathogenic role of fibronectin in RCC.

## 4. Materials and Methods

### 4.1. Cell Cultures

786-O (ATCC CRL1932) and Caki-1 (ATCC HTB-46) cells were purchased from American Type Culture Collection (Rockville, MD, USA) and propagated in Dulbecco’s modified Eagle medium (DMEM) with 10% fetal bovine serum (FBS) by placing cells in an incubator with 5% CO_2_ at 37 °C [11,34]. To evaluate the effects of hypoxia, cells were placed in an incubator with 1% O_2_, 5% CO_2_, and 94% N_2_ for 8 h [9].

### 4.2. Cell Growth

For the measurement of cell growth, cells were seeded onto 96-well plates at a density of 1 × 10^4^ per well. During experiments, the CellTiter 96^®^ AQ_ueous_ One Solution Cell Proliferation Assay (Promega, Madison, WI, USA) was used to assess cell growth according to the manufacturer’s instructions.

### 4.3. Colony Formation Assay

Cells were seeded onto six-well plates at a density of 250 per well and maintained in DMEM containing 10% FBS for seven days. Afterwards, cells were fixed and cell colony was visualized by staining with crystal violet [35].

### 4.4. Wound Healing Assay

Cells were seeded onto six-well plates at a density of 4 × 10^5^/mL and grown until confluence was reached. The confluent cell monolayer was then scratched with a tip of a 200-μL micropipette to create a wound and maintained in DMEM containing 0.5% FBS. Photomicrographs were taken immediately after the scraping and 16 h later. The areas of gaps without cell coverage were measured and the percentage of wound closure was calculated [9].

### 4.5. Cell Migration Assay

Cell migration was measured using a 24-well Transwell permeable chamber with 8 μm pore size (BD Falcon Cell Culture insert, BD Biosciences, San Jose, CA, USA). Briefly, cell suspension (2 × 10^4^) in 200 μL DMEM containing 2% FBS was added to the top of the Transwell inserts without or with fibronectin coating (50 μg/mL), and 600 μL DMEM containing 10% FBS was put into the bottom well for 24 h. Furthermore, the upper wells and lower wells were replaced with DMEM containing 0.1% bovine serum albumin (BSA) and DMEM containing 0.1% BSA and fibronectin (50 μg/mL), respectively, in some studies and antibody neutralization assay. For the measurement of cell membrane receptors involved in fibronectin-mediated migration, 786-O cells were first incubated with isotype control IgG (5 μg/mL, 12-371, Millipore, Burlington, MA, USA), integrin β1 IgG (5 μg/mL, MAB1959, Millipore, Burlington, MA, USA), or integrin α5 IgG (5 μg/mL, MAB1956, Millipore, Burlington, MA, USA) for 30 min before seeding to the Transwell inserts for migration assay. The migrating cells in the lower surfaces of the Transwell inserts were stained with Giemsa. For statistical comparison, six random fields per insert (10×) were selected for measurement [9].

### 4.6. Small Interfering RNA (siRNA) Transfection

The siRNA against human fibronectin (sc-29315), integrin α5 (sc-29372), integrin β1 (sc-35674), and control siRNA (sc-37007) were purchased from Santa Cruz Biotechnology (Santa Cruz, CA, USA). The delivery of siRNA into the cells was conducted using INTERFERinTM siRNA transfection reagent (Polyplus-transfection, Inc., New York, NY, USA) in accordance with the manufacturer’s instructions [35].

### 4.7. Enzyme-Linked Immunosorbent Assay (ELISA)

The levels of TGF-β1 in supernatants were measured using an ELISA kit according to the manufacturer’s instructions (R&D Systems, Minneapolis, MN, USA).

### 4.8. Western Blot

Cells were lysed in ice-cold Laemmli sodium dodecyl sulfate (SDS) sample buffer. Equal amounts of proteins were separated by 10% or 12% SDS-PAGE and transferred onto PVDF membranes which were incubated with specific antibodies followed by horseradish peroxidase-labeled IgG. All antibodies were reacted with fibronectin, integrin α5, integrin β1, FAK, phospho-FAK (Tyr-397), ERK, phospho-ERK (Thr-202/Tyr-204), Akt, phospho-Akt (Ser-473), Src, phospho-Src (Tyr-416), cyclin D1, vimentin, Smad (Santa Cruz Biotechnology, Santa Cruz, CA, USA), phospho-Smad (Ser-423/425, Cell Signaling, Danvers, MA, USA), or glyceraldehyde-3-phosphate dehydrogenase (GAPDH) (R&D Systems, Minneapolis, MN, USA). The signals in membranes were visualized using enhanced chemiluminescence (ECL) Western blotting reagents and the intensity of signals was determined by a computer image analysis system (Alpha Innotech Corporation, IS1000, San Leandro, CA, USA).

### 4.9. Statistical Analysis

All data are expressed as mean values ± standard deviation. Statistical significance between groups was carried out using one-way analysis of variance followed by Student’s *t*-test. A level of *p* < 0.05 was considered statistically significant.

## Figures and Tables

**Figure 1 ijms-20-02792-f001:**
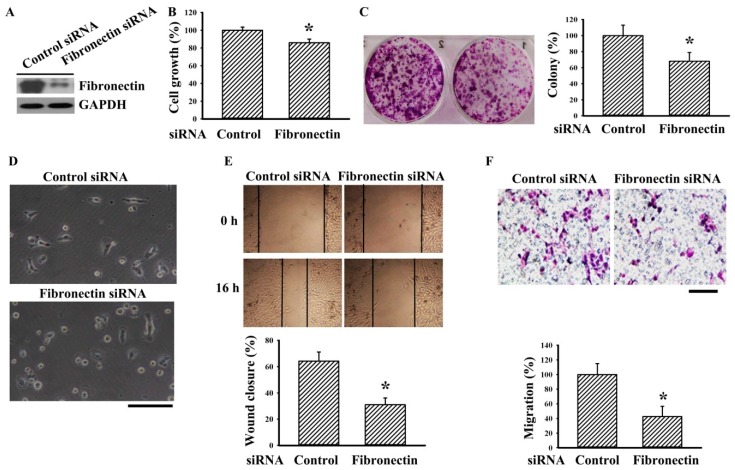
Fibronectin silencing decreased cell growth and migration in 786-O cells. 786-O cells were transfected with control siRNA and fibronectin siRNA for 48 h. (**A**) Proteins were extracted and subjected to Western blot analysis with indicated antibodies. Representative blots are shown. (**B**) The resultant transfected cells were seeded onto 96-well plates. Twenty-four hours later, cell growth was measured by MTS reduction assay. (**C**) The resultant transfected cells were seeded onto six-well plates for seven days. Cell colonies were fixed and stained with crystal violet. Representative plates are shown. The numbers of cell colonies were calculated and depicted. (**D**) The resultant transfected cells were seeded onto six-well plates for 1 h. Cells were examined under a light microscope. Representative photomicrographs are shown. Scale bar = 50 μm. (**E**) The resultant transfected cells were seeded onto six-well plates for 24 h. When confluence was reached, cell movement was evaluated by a wound-healing assay for 16 h in the presence of 0.5% FBS. Representative photomicrographs are shown. Bar graphs show relative wound closure among groups. (**F**) The resultant transfected cells were seeded onto Transwell inserts and subjected to Transwell migration assay for 24 h. The lower chambers were filled with DMEM containing 10% FBS. Representative photomicrographs are shown. Scale bar = 50 μm. Bar graphs show quantitative results among groups and the value in the control siRNA group was defined as 100% (**B**, **C**, and **F**). * *p* < 0.05 vs. control siRNA group, *n* = 3.

**Figure 2 ijms-20-02792-f002:**
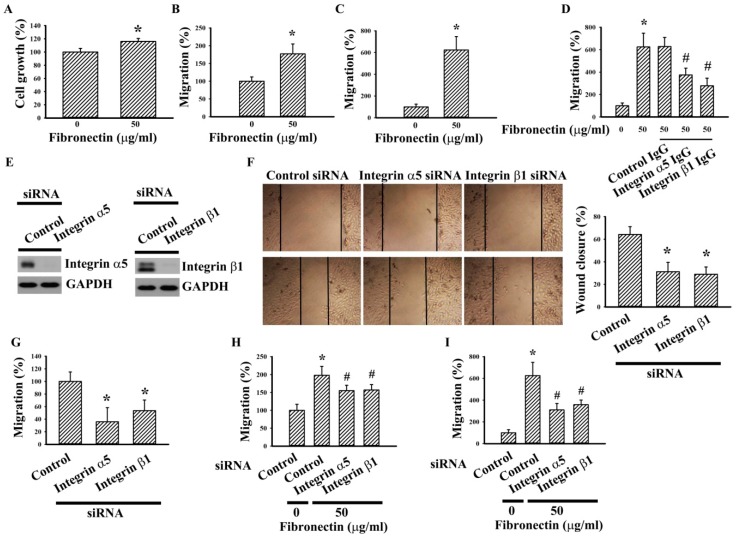
Integrin α5 and integrin β1 silencing alleviated fibronectin effects in 786-O cells. (**A**) 786-O cells were seeded onto fibronectin (0 and 50 μg/mL)-coated 96-well plates. Twenty-four hours later, cell growth was measured by MTS reduction assay. (**B**) 786-O cells were seeded onto fibronectin (0 and 50 μg/mL)-coated Transwell inserts and subjected to Transwell migration assay for 24 h. The lower chambers were filled with DMEM containing 10% FBS. (**C**) 786-O cells were seeded onto Transwell inserts and subjected to Transwell migration assay for 24 h. The lower chambers were filled with DMEM containing fibronectin (0 or 50 μg/mL). (**D**) 786-O cells were first incubated with indicated IgG (5 μg/mL) for 30 min before seeding to the Transwell inserts for migration assay (24 h). The lower chambers were filled with DMEM containing fibronectin (0 or 50 μg/mL). (**E**) 786-O cells were transfected with control siRNA, integrin α5 siRNA, and integrin β1 siRNA for 48 h. Proteins were extracted and subjected to Western blot analysis with indicated antibodies. Representative blots are shown. (**F**) The resultant transfected cells were seeded onto six-well plates for 24 h. When confluence was reached, cell movement was evaluated by a wound-healing assay for 16 h in the presence of 0.5% FBS. Representative photomicrographs are shown. Bar graphs showed relative wound closure among groups. (**G**) The resultant transfected cells were seeded onto Transwell inserts and subjected to Transwell migration assay for 24 h. The lower chambers were filled with DMEM containing 10% FBS. (**H**) The resultant transfected cells were seeded onto fibronectin (0 and 50 μg/mL)-coated Transwell inserts and subjected to Transwell migration assay for 24 h. The lower chambers were filled with DMEM containing 10% FBS. (**I**) The resultant transfected cells were seeded onto Transwell inserts and subjected to Transwell migration assay for 24 h. The lower chambers were filled with DMEM containing fibronectin (0 and 50 μg/mL). Bar graphs show quantitative results among groups and the value in fibronectin (0 μg/mL)/control siRNA group was defined as 100% (**A**–**D**, **G**–**I**). * *p* < 0.05 vs. fibronectin (0 μg/mL)/control siRNA group and # *p* < 0.05 vs. fibronectin (50 μg/mL)/control siRNA group, *n* = 3.

**Figure 3 ijms-20-02792-f003:**
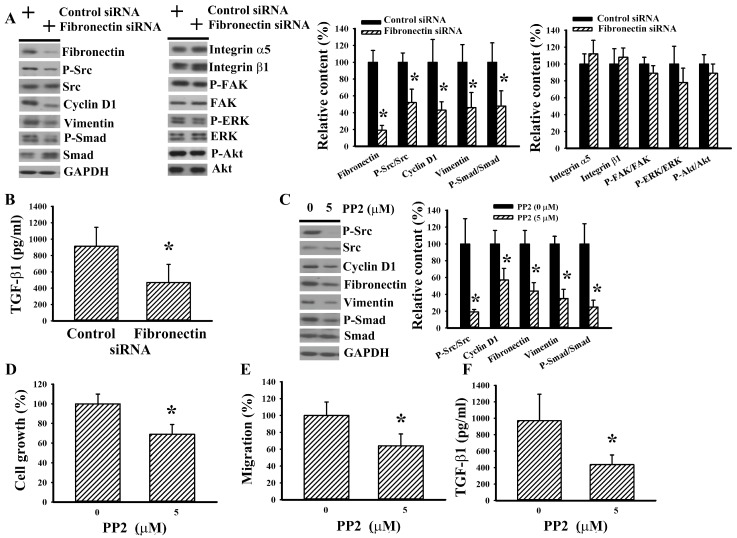
Fibronectin silencing decreased intracellular Src signaling in 786-O cells. 786-O cells were transfected with control siRNA and fibronectin siRNA for 48 h. (**A**) Proteins were extracted and subjected to Western blot analysis with indicated antibodies. Representative blots are shown. (**B**) The resultant transfected cells were seeded onto 24-well plates. Twenty-four hours later, the supernatants were collected and subjected to ELISA for the measurement of TGF-β1. (**C**) 786-O cells were treated with PP2 (0 and 5 μM) for 5 h. Proteins were extracted and subjected to Western blot analysis with indicated antibodies. Representative blots are shown. (**D**) 786-O cells were treated with PP2 (0 and 5 μM) for 24 h. Cell growth was measured by MTS reduction assay. (**E**) 786-O cells were seeded onto Transwell inserts and subjected to Transwell migration assay for 24 h in the presence of PP2 (0 and 5 μM). The lower chambers were filled with DMEM containing 10% FBS. (**F**) 786-O cells were treated with PP2 (0 and 5 μM) for 24 h. The supernatants were collected and subjected to ELISA for the measurement of TGF-β1. Bar graphs show quantitative results among groups and the value in control siRNA/untreated group was defined as 100% (**A**, **C**–**E**). * *p* < 0.05 vs. control siRNA/untreated group, *n* = 3.

**Figure 4 ijms-20-02792-f004:**
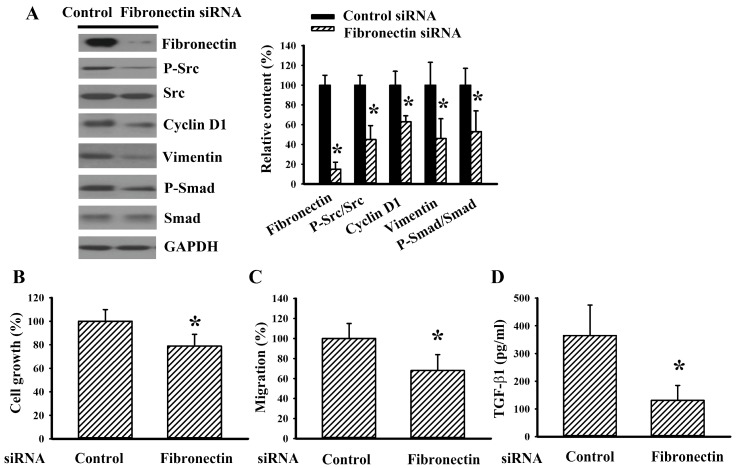
Fibronectin silencing decreased cell growth and migration in Caki-1 cells. Caki-1 cells were transfected with control siRNA and fibronectin siRNA for 48 h. (**A**) Proteins were extracted and subjected to Western blot analysis with indicated antibodies. Representative blots are shown. (**B**) The resultant transfected cells were seeded onto 96-well plates. Twenty-four hours later, cell growth was measured by MTS reduction assay. (**C**) The resultant transfected cells were seeded onto Transwell inserts and subjected to Transwell migration assay for 24 h. The lower chambers were filled with DMEM containing 10% FBS. (**D**) The resultant transfected cells were seeded onto 24-well plates. Twenty-four hours later, the supernatants were collected and subjected to ELISA for the measurement of TGF-β1. Bar graphs showed quantitative results among groups and the value in control siRNA group was defined as 100% (**A**–**C**). * *p* < 0.05 vs. control siRNA group, *n* = 3.

**Figure 5 ijms-20-02792-f005:**
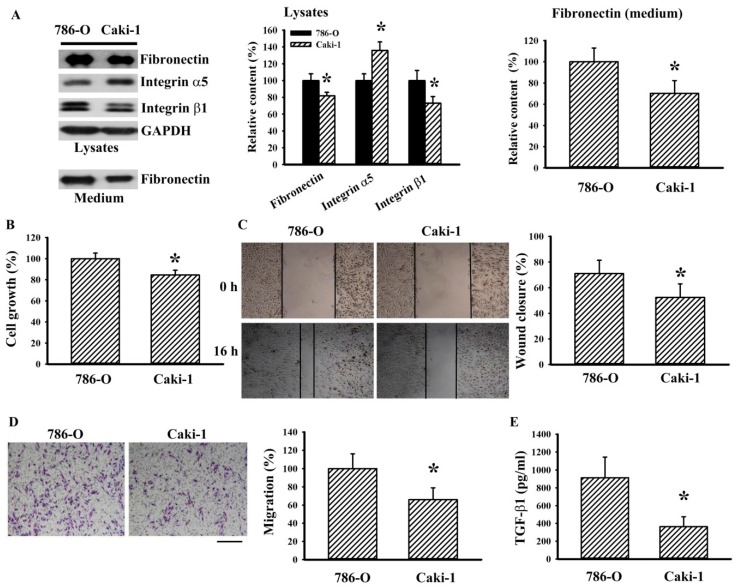
Cell growth and migration potential of RCC cells were associated with fibronectin. (**A**) 786-O and Caki-1 cells were seeded onto six-well plates and cultured for 24 h. Proteins were extracted from subconfluent (80%) cells (Lysates) and supernatants (Medium, 20 μL) and subjected to Western blot analysis with indicated antibodies. Representative blots are shown. (**B**) 786-O and Caki-1 cells were seeded onto 96-well plates. Twenty-four hours later, cell growth was measured by MTS reduction assay. (**C**) 786-O and Caki-1 cells were seeded onto six-well plates for 24 h. When reaching confluence, cell movement was evaluated by a wound-healing assay for 16 h in the presence of 0.5% FBS. Representative photomicrographs are shown. Bar graphs show relative wound closure among groups. (**D**). 786-O and Caki-1 cells were seeded onto Transwell inserts and subjected to Transwell migration assay for 24 h. The lower chambers were filled with DMEM containing 10% FBS. Representative photomicrographs are shown. Scale bar = 100 μm. (**E**) 786-O and Caki-1 cells were seeded onto 24-well plates. Twenty-four hours later, the supernatants were collected and subjected to ELISA for the measurement of TGF-β1. Bar graphs show quantitative results among groups and the value in 786-O group was defined as 100% (**A**,**B**,**D**). * *p* < 0.05 vs. 786-O group, *n* = 3.

**Figure 6 ijms-20-02792-f006:**
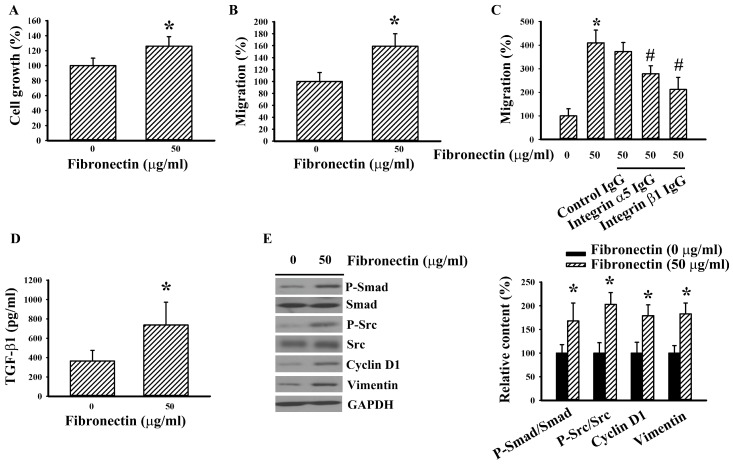
Fibronectin increased cell growth and migration in Caki-1 cells. (**A**) Caki-1 cells were seeded onto fibronectin (0 and 50 μg/mL)-coated 96-well plates. Twenty-four hours later, cell growth was measured by MTS reduction assay. (**B**) Caki-1 cells were seeded onto fibronectin (0 and 50 μg/mL)-coated Transwell inserts and subjected to Transwell migration assay for 24 h. The lower chambers were filled with DMEM containing 10% FBS. (**C**) Caki-1 cells were first incubated with indicated IgG (5 μg/mL) for 30 min before seeding to the Transwell inserts for migration assay (24 h). The lower chambers were filled with DMEM containing fibronectin (0 or 50 μg/mL). (**D**) Caki-1 cells were seeded onto fibronectin (0 and 50 μg/mL)-coated 24-well plates. Twenty-four hours later, the supernatants were collected and subjected to ELISA for the measurement of TGF-β1. (**E**) Caki-1 cells were seeded onto fibronectin (0 and 50 μg/mL)-coated six-well plates for 5 h. Proteins were extracted and subjected to Western blot analysis with indicated antibodies. Representative blots are shown. Bar graphs show quantitative results among groups and the value in uncoated group was defined as 100% (**A**–**C**, and **E**). * *p* < 0.05 vs. uncoated control and # *p* < 0.05 vs. fibronectin (50 μg/mL) untreated control, *n* = 3.

**Figure 7 ijms-20-02792-f007:**
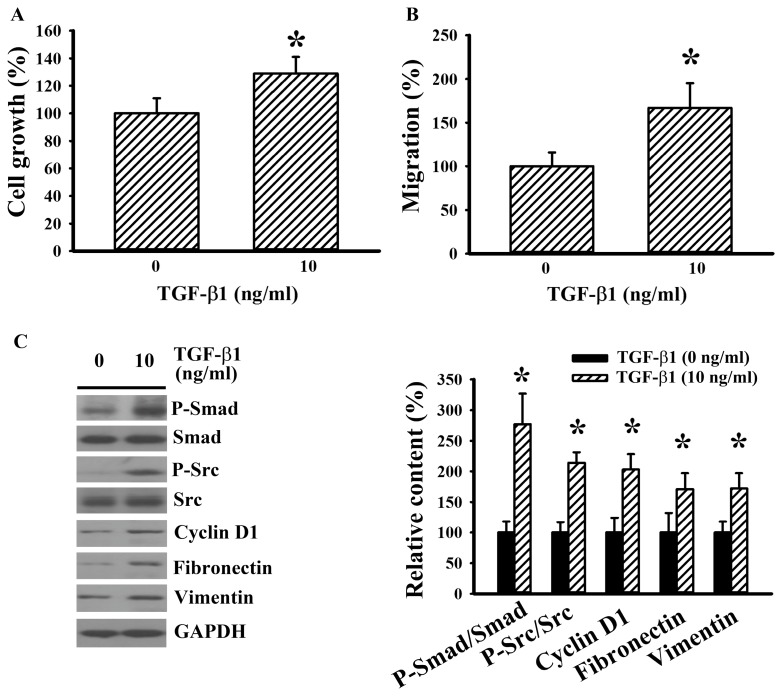
TGF-β1 increased cell growth and migration in Caki-1 cells. (**A**) Caki-1 cells were treated with TGF-β1 (0 and 10 ng/mL) for 24 h. Cell growth was measured by MTS reduction assay. (**B**) Caki-1 cells were seeded onto Transwell inserts and subjected to Transwell migration assay for 24 h in the presence of TGF-β1 (0 and 10 ng/mL). The lower chambers were filled with DMEM containing 10% FBS. (**C**) Caki-1 cells were treated with TGF-β1 (0 and 10 ng/mL) for 5 h. Proteins were extracted and subjected to Western blot analysis with indicated antibodies. Representative blots are shown. Bar graphs show quantitative results among groups and the value in untreated group was defined as 100% (**A**–**C**). * *p* < 0.05 vs. untreated control, *n* = 3.

**Figure 8 ijms-20-02792-f008:**
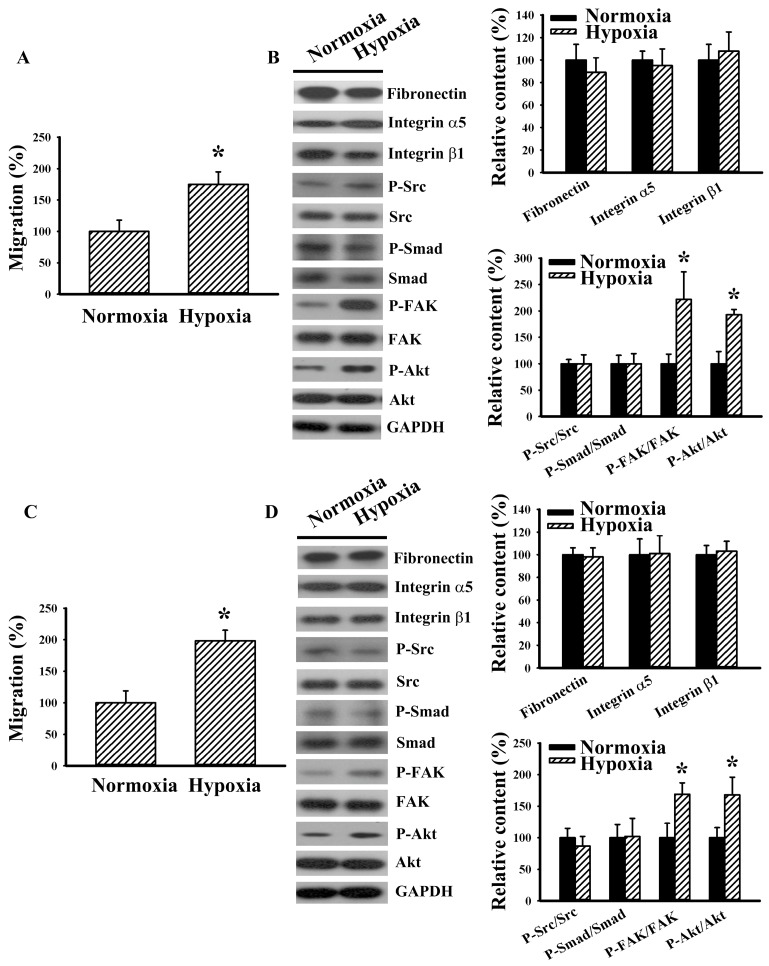
Hypoxia increased RCC cell migration. (**A**) 786-O cells were seeded onto Transwell inserts and subjected to Transwell migration assay in a normoxia or hypoxia incubator (1% O_2_, 5% CO_2_, and 94% N_2_) for 8 h. The lower chambers were filled with DMEM containing 10% FBS. (**B**) 786-O cells were placed in a normoxia or hypoxia incubator (1% O_2_, 5% CO_2_, and 94% N_2_) for 8 h. Proteins were extracted and subjected to Western blot analysis with indicated antibodies. Representative blots are shown. (**C**) Caki-1 cells were seeded onto Transwell inserts and subjected to Transwell migration assay in a normoxia or hypoxia incubator (1% O_2_, 5% CO_2_, and 94% N_2_) for 8 h. The lower chambers were filled with DMEM containing 10% FBS. (**D**) Caki-1 cells were placed in a normoxia or hypoxia incubator (1% O_2_, 5% CO_2_, and 94% N_2_) for 8 h. Proteins were extracted and subjected to Western blot analysis with indicated antibodies. Representative blots are shown. Bar graphs show quantitative results among groups and the value in normoxia group was defined as 100% (**A**–**D**). * *p* < 0.05 vs. normoxia control, *n* = 3.

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
