# Peer review of "Fibronectin Promotes Cell Growth and Migration in Human Renal Cell Carcinoma Cells"

_ijms, 2019, doi:10.3390/ijms20112792_

Reviewer 1 Report

The article entitled " Fibronectin Promotes Proliferation and Migration in Human Renal Cell Carcinoma Cells "highlight the role of fibronectin in RCC proliferation and migration involving mutual actions with Src and TGF-b1.

The article is well written and well organized. Some points could be improved to reinforce the paper.

 Results

2.1. 

Figure 1D is too small to observe cells morphology.

Figure 1E  Concerning wound healing and transwell experiments.  It ‘s important to precise the part of the effect linked to proliferation. 

A possibility is to inhibit mitosis by mitomycin C pretreatment (Oncogene volume 35pages 2687–2697 (2016))

2.2 Authors have to precise why they focused only on α5β1integrin and not αvβ3or α4β1.

A screening of these integrins could be added and/or an experiment of siRNA on αv, α4, β3 to justify others experiments on α5β1. Is it an RGD dependent effect?

Figure 2A

In the different experiments (other figures), “viability” should be changed by “cell growth” because there is no measurement of cell death vs alive cells (trypan blue exclusion).

2.3

Figure 3. Authors have studied the effect of fibronectin silencing on intracellular signaling but they don’t do it with integrins silenced cells. If they do it only with fibronectin and if they don’t precise integrin status, others integrins could be involved in the signalization. Then, it is difficult to link the first results with these ones. 

Figure 3. Phosphorylation residues have to be precised.

2.4

Authors used a new cell line to go on their experiments. This use has to be justify (integrin status, fibronectin synthesis…). A screening of various kidney cell lines should be added.

2.5

Authors indicated that “Proliferation and migration potential of RCC cells were associated with fibronectin”. 

In this part, the wound after 16h for 786-O cells (Figure 5C) is different from wound after 16h in figure 2E. Why?

2.6.

Authors suggest potential crosstalk between fibronectin, Src, and TGF. As TGF acts on various signaling pathways, authors have to precise this effect.

What is the effect of TGFB1 on integrin expression.  Is there an effect of TGB1 on cell migration (is this effet is different when we use control or SiRNA silenced cells, RGD peptide?).

Author Response

First, we thank deeply to those constructive and instructive comments from the reviewers and editors. We sincerely considered these comments and made an appropriate improvement of this revised manuscript (ijms-500418) within 10 days. As suggested by the reviewers, the figures were re-drawn in a friendly way. Besides, data of antibody neutralization study, protein expression between 786-O and Caki-1 cells, and hypoxia-promoted cell migration were added to the revised version manuscript. We also apologize to the reviewers because that some experiments were not added due to the limited revision time and materials. The detailed summary of the changes in relation to the reviewers was listed as follows and marked as red color in the revised version of manuscript.

Comments and Suggestions for Authors

The article entitled "Fibronectin Promotes Proliferation and Migration in Human Renal Cell Carcinoma Cells"highlight the role of fibronectin in RCC proliferation and migration involving mutual actions with Src and TGF-b1. The article is well written and well organized. Some points could be improved to reinforce the paper.

Results

2.1.

Figure 1D is too small to observe cells morphology.

Responses:

As suggested by the reviewer, figures were enlarged.

Figure 1E Concerning wound healing and transwell experiments. It‘s important to precise the part of the effect linked to proliferation. A possibility is to inhibit mitosis by mitomycin C pretreatment (Oncogene volume 35, pages 2687-2697 (2016)).

Responses:

The wound healing assay was conducted by placing cells in medium containing 0.5% FBS to minimize cell proliferation. However, the effect of cell proliferation could not be totally excluded. Related descriptions of the concern were added. Please refer to line 305-311, line 394-399.

2.2 Authors have to precise why they focused only on α5β1integrin and not αvβ3 or α4β1. A screening of these integrins could be added and/or an experiment of siRNA on αv, α4, β3 to justify others experiments on α5β1. Is it an RGD dependent effect?

Responses:

Integrin a5 and integrin b1 are common dimeric receptor of fibronectin. In the beginning, we first tested the potential involvement of integrin a5 and integrin b1 in RCC cell migration. Other integrin members were not characterized. Due to the limited revision time, antibodies and siRNAs of other integrins were not afforded. We added antibody neutralization assay to further indicate the involvement of integrin a5 and integrin b1 (Figure 2D and Figure 6C). Related descriptions of the concern were added. Please refer to line 111-118, line 293-311.

Figure 2A

In the different experiments (other figures), “viability” should be changed by “cell growth” because there is no measurement of cell death vs alive cells (trypan blue exclusion).

Responses:

As suggested by the reviewer, the term was changed to cell growth. Besides, the title was also changed to “Fibronectin Promotes Cell Growth and Migration in Human Renal Cell Carcinoma Cells”.

2.3

Figure 3. Authors have studied the effect of fibronectin silencing on intracellular signaling but they don’t do it with integrins silenced cells. If they do it only with fibronectin and if they don’t precise integrin status, others integrins could be involved in the signalization. Then, it is difficult to link the first results with these ones.

Responses:

We added antibody neutralization assay to further indicate the involvement of integrin a5 and integrin b1 (Figure 2D and Figure 6C). Related descriptions of the concern and study limitation were added. Please refer to line 111-118, line 293-311, line 343-354.

Figure 3. Phosphorylation residues have to be precised.

Responses:

As suggested by the reviewer, the phosphorylated amino acid residue was added. Please refer to line 428-438.

2.4

Authors used a new cell line to go on their experiments. This use has to be justify (integrin status, fibronectin synthesis…). A screening of various kidney cell lines should be added.

Responses:

As suggested by the reviewer, the expression levels of fibronectin, integrin a5, and integrin b1 were compared between 786-O and Caki-1 cells (Figure 5A). Related descriptions of the concern were added. Please refer to line 192-200, line 343-354.

2.5

Authors indicated that “Proliferation and migration potential of RCC cells were associated with fibronectin”. In this part, the wound after 16h for 786-O cells (Figure 5C) is different from wound after 16h in figure 2E. Why?

Responses:

Each study was done by three batches of cells. Experiments of Figure 2F and Figure 5C were conducted separately. Representative photographs were shown.

2.6.

Authors suggest potential crosstalk between fibronectin, Src, and TGF. As TGF acts on various signaling pathways, authors have to precise this effect. What is the effect of TGFB1 on integrin expression. Is there an effect of TGB1 on cell migration (is this effect is different when we use control or SiRNA silenced cells, RGD peptide?).

Responses:

In this study, silencing of fibronectin decreased and exogenous fibronectin increased the expression of Src and TGF-b1/Smad signaling. Likewise, the addition of TGF-b1 promoted cell growth, migration, fibronectin expression, and Src phosphorylation. Src inhibitor decreased cell growth, migration, TGF-b1 production, and Smad phosphorylation. Our findings suggest that the action of fibronectin involves the Src and TGF-b1/Smad signaling. Related descriptions of the concern were added. Please refer to line 217-227, line 323-341.

Reviewer 2 Report

The paper presents the study of the evaluation of pro-migratory and pro-proliferative effect of fibronectin in two different human renal cell carcinoma cell lines: 786-O and Caki-1 cells. The manuscript is well written and constructed. The experiments  are performed in a proper way, however, they should be presented  in a more “friendly” way (increased size of photos from WB, colony formation or wound healing assay, increased fonts)

The only one concern regarding the experiments is that they were performed only in normoxic conditions, whereas it is well known that hypoxia influences the progression of renal cell carcinoma. What will be the effect of low oxygen concentration on  studied parameters? At least discussion should contain some considerations about the influence of hypoxia.

I have also one comment about the methods used. The assay called scratch assay or wound healing assay measures in fact proliferation and migration (or only migration if proliferation is blocked by e.g. hydroxyurea). Therefore, in my opinion, it is overestimated to write that e.g. “Silencing of integrin α5 and integrin β1 also impaired 786-O cell wound healing and migration as well as diminished the pro-migratory and pro-chemotactic effects of fibronectin” (Abstract) or “Fibronectin-silenced 786-O cells showed lower effectiveness in wound healing” (page 2, line 86). Please correct those sentences and underline the decrease in proliferation and migration (as the Authors did not block proliferation).

Minor point:

Please correct – Samd for Smad - Page 9 line 260  - Crosstalk among fibronectin, Src, and TGF-β1/Samd

Author Response

First, we thank deeply to those constructive and instructive comments from the reviewers and editors. We sincerely considered these comments and made an appropriate improvement of this revised manuscript (ijms-500418) within 10 days. As suggested by the reviewers, the figures were re-drawn in a friendly way. Besides, data of antibody neutralization study, protein expression between 786-O and Caki-1 cells, and hypoxia-promoted cell migration were added to the revised version manuscript. We also apologize to the reviewers because that some experiments were not added due to the limited revision time and materials. The detailed summary of the changes in relation to the reviewers was listed as follows and marked as red color in the revised version of manuscript.

Comments and Suggestions for Authors

The paper presents the study of the evaluation of pro-migratory and pro-proliferative effect of fibronectin in two different human renal cell carcinoma cell lines: 786-O and Caki-1 cells. The manuscript is well written and constructed. The experiments are performed in a proper way, however, they should be presented in a more “friendly” way (increased size of photos from WB, colony formation or wound healing assay, increased fonts)

Responses:

As suggested by the reviewer, the figures were re-drawn and enlarged.

The only one concern regarding the experiments is that they were performed only in normoxic conditions, whereas it is well known that hypoxia influences the progression of renal cell carcinoma. What will be the effect of low oxygen concentration on studied parameters? At least discussion should contain some considerations about the influence of hypoxia.

Responses:

As suggested by the reviewer, the effects of hypoxia on cell migration were conducted in 786-O and Caki-1 cells after a 8-h of hypoxia. We found that hypoxia promoted cell migration accompanied by increased FAK and Akt phosphorylation with the exception of fibronectin, integrin a5, integrin b1, Src phosphorylation and Smad phosphorylation (Figure 8). Related descriptions of the concern were added. Please refer to line 252-261, line 264-274, line 355-366, line 378-382.

I have also one comment about the methods used. The assay called scratch assay or wound healing assay measures in fact proliferation and migration (or only migration if proliferation is blocked by e.g. hydroxyurea). Therefore, in my opinion, it is overestimated to write that e.g. “Silencing of integrin α5 and integrin β1 also impaired 786-O cell wound healing and migration as well as diminished the pro-migratory and pro-chemotactic effects of fibronectin” (Abstract) or “Fibronectin-silenced 786-O cells showed lower effectiveness in wound healing” (page 2, line 86). Please correct those sentences and underline the decrease in proliferation and migration (as the Authors did not block proliferation).

Responses:

The wound healing assay was conducted by placing cells in medium containing 0.5% FBS to minimize cell proliferation. However, the effect of cell proliferation could not be totally excluded. Related descriptions of the concern were added. Please refer to line 27-29, line 84-85, line 293-311.

Minor point:

Please correct – Samd for Smad - Page 9 line 260 - Crosstalk among fibronectin, Src, and TGF-β1/Samd

Responses:

As suggested by the reviewer, the error was corrected.

Round  2

Reviewer 1 Report

Authors have made most of the asked corrections. 

Minor point could still be corrected

“We added antibody neutralization assay to further indicate the involvement of integrin a5 and integrin b1 (Figure 2D and Figure 6C). Related descriptions of the concern and study limitation were added. Please refer to line 111-118, line 293-311, line 343-354. »

In fact, we can observe a better effect with IgG anti-B1 than A5. An explication is probably that an other integrin than A5 involved. This point should be discussed in the discussion.

Author Response

First, we thank deeply to those constructive and instructive comments from the reviewers and editors. We sincerely considered these comments and made an appropriate improvement of this revised manuscript (ijms-500418). The detailed summary of the changes in relation to the reviewers was listed as follows and marked as red color in the revised version of manuscript.

Comments and Suggestions for Authors

Authors have made most of the asked corrections.

Minor point could still be corrected

“We added antibody neutralization assay to further indicate the involvement of integrin a5 and integrin b1 (Figure 2D and Figure 6C). Related descriptions of the concern and study limitation were added. Please refer to line 111-118, line 293-311, line 343-354.

In fact, we can observe a better effect with IgG anti-b1 than a5. An explication is probably that an other integrin than a5 involved. This point should be discussed in the discussion.

Responses:

According to data of antibody neutralization assay, neutralization of integrin b1 produced a better effect than integrin a5. However, fibronectin-induced chemotactic migration was not totally inhibited by integrin b1 or integrin a5 neutralization. The findings indicated that other integrin receptor subunits should be taken into consideration. 786-O cells showed faster migration than Caki-1 cells, accompanied by higher expression of integrin b1 and lower expression of integrin a5. It means that integrin b1 might be one cause for the discrepancy.

Related descriptions of the concern were added. Please refer to line 293-315, line 347-361.